# An Overview of Potential Natural Photosensitizers in Cancer Photodynamic Therapy

**DOI:** 10.3390/biomedicines11010224

**Published:** 2023-01-16

**Authors:** Bushra Aziz, Iffat Aziz, Ahmat Khurshid, Ehsan Raoufi, Fahime Nasr Esfahani, Zahra Jalilian, M. R. Mozafari, Elham Taghavi, Masroor Ikram

**Affiliations:** 1Biophotonics and Photonanomedicine Research Laboratory, Department of Physics and Applied Mathematics, Pakistan Institute of Engineering and Applied Sciences (PIEAS), Nilore, Islamabad 45650, Pakistan; 2Department of Physics, Women University of Azad Jammu & Kashmir, Bagh, Azad Kashmir 12500, Pakistan; 3The National Metrology Institute of Pakistan, Islamabad 44800, Pakistan; 4Australasian Nanoscience and Nanotechnology Initiative (ANNI), Monash University LPO, Clayton, VIC 3168, Australia; 5Faculty of Fisheries and Food Science, Universiti Malaysia Terengganu, Kuala Nerus 21030, Terengganu, Malaysia

**Keywords:** photodynamic therapy, photosensitizer, plant extract, medicinal plants, cancer, ROS

## Abstract

Cancer is one of the main causes of death worldwide. There are several different types of cancer recognized thus far, which can be treated by different approaches including surgery, radiotherapy, chemotherapy or a combination thereof. However, these approaches have certain drawbacks and limitations. Photodynamic therapy (PDT) is regarded as an alternative noninvasive approach for cancer treatment based on the generation of toxic oxygen (known as reactive oxygen species (ROS)) at the treatment site. PDT requires photoactivation by a photosensitizer (PS) at a specific wavelength (λ) of light in the vicinity of molecular oxygen (singlet oxygen). The cell death mechanisms adopted in PDT upon PS photoactivation are necrosis, apoptosis and stimulation of the immune system. Over the past few decades, the use of natural compounds as a photoactive agent for the selective eradication of neoplastic lesions has attracted researchers’ attention. Many reviews have focused on the PS cell death mode of action and photonanomedicine approaches for PDT, while limited attention has been paid to the photoactivation of phytocompounds. Photoactivation is ever-present in nature and also found in natural plant compounds. The availability of various laser light setups can play a vital role in the discovery of photoactive phytocompounds that can be used as a natural PS. Exploring phytocompounds for their photoactive properties could reveal novel natural compounds that can be used as a PS in future pharmaceutical research. In this review, we highlight the current research regarding several photoactive phytocompound classes (furanocoumarins, alkaloids, poly-acetylenes and thiophenes, curcumins, flavonoids, anthraquinones, and natural extracts) and their photoactive potential to encourage researchers to focus on studies of natural agents and their use as a potent PS to enhance the efficiency of PDT.

## 1. Introduction

The word cancer describes a collection of diseases that arise due to abnormalities/changes in the genes (DNA mutation). It is a leading cause of death worldwide and it has a very high impact on global health [1]. Abnormal cell growth due to genetic mutation causes the development of a tumor that is either benign, malignant or non-malignant [2].

Different types of therapeutic modalities such as radiotherapy, chemotherapy, surgery, or combinations of these are used to combat this life-threatening disease. However, each of these approaches has its own limitations, e.g., invasiveness, suppression of the immune system, long-term side effects, dose-dependent side effects, etc. [3]. So, there is still a need for additional innovative approaches to combat cancers. Photodynamic therapy (PDT) is regarded as an alternative low-level invasive approach to cancer treatment based on the generation of reactive oxygen species (ROS) and free radicals at the treatment site. As compared to radiotherapy, chemotherapy, and surgery its side effects are minimal and more tolerable due to its selective activation of drugs [4].

In PDT, a particular type of drug (photo-activated) is used, which is only activated by exposure to a specific wavelength of light in the presence of cellular molecular oxygen (O_2_), causing it to become toxic to the cancer cell. Without illumination, this drug is nontoxic for cells. This is one of the most appealing features of PDT. This particular type of drug is called a photosensitizer (PS) [5]. A PS accumulates for a longer time in a cancer cell as compared to a normal cell. This prolonged retention in the neoplasm region is due to poor lymphatic drainage and increased blood vessel permeability [6]. Over the past few decades, different types of synthetic inorganic and organic PSs have been explored and reported. Among these, some have been translated into clinical trials [7]. There are three different inter-related fundamental cell death pathways:(1)Pathways that directly kill the cancer cell;(2)Those that damage the vasculature to stop the oxygen supply to cells;(3)Those that activate/stimulate the systemic immunity response.

PDT incorporates these three pathways to kill the cancer cells [8,9].

Since very ancient times, herbal medicines have been used to treat various human cancerous [10,11] and non-cancerous [12,13] diseases. Over the past few decades, the research has focused on screening herbal substances (medicinal plants) to determine their chemo-therapeutic properties [14,15] since they are environmentally sustainable and have minimal side effects [16]. However, many active pharmaceutical anti-cancer agents that are isolated from medicinal natural plants have not been screened for photoactive properties. Nature provides a valuable reservoir of medicinal plants (a source of natural compounds) that have the potential to be synthesized and used as pharmaceutical anticancer agents [15,17,18]. Few studies have attempted to identify new photoactive chemical compounds from medicinal plant extracts that could be potent new natural PSs [19,20,21,22]. Research on natural compounds has demonstrated that a few photoactive agents in various medicinal plants are as efficient as traditional PSs [23]. These studies suggest that natural compounds with photosensitizing capabilities can be isolated from natural extracts and have the potential to be employed as an alternative to the conventional photosensitizers used in PDT.

In this review article, a brief fundamental philosophy of PDT, conventional PSs, FDA approved PSs, and natural light-activable compounds are discussed. This report is primarily focused on the anticancer and phototoxic activity of plant-based natural compounds (furanocoumarins, alkaloids, poly-acetylenes and thiophenes, curcumins, flavonoids, anthraquinones, and natural extracts).

## 2. Basics of Photodynamic Therapy (PDT)

Photodynamic therapy (PDT) is based on the complex coordination of the following three factors:(1)The specific wavelength of light;(2)Cellular oxygen; and(3)A light-activated drug, namely, a photosensitizer, as depicted in Figure 1 [4].

PDT fails if any of these three factors are missing. These three factors are individually harmless or non-toxic to the cells but when synchronized with each other, they become toxic to the cancer cells by activating a photochemical reaction.

As presented in Figure 1, when the PS at a ground state (S0) is irradiated by a specific wavelength of light, then it is activated and jumps up to an unstable singlet excited state (S2), with a lifetime of a nano-second (10^−9^ s). It quickly takes advantage of internal conversion (IC) to jump down to a lower unstable singlet excited state (S1), with a short lifetime duration (10^−9^ s). Then it quickly jumps to a more stable excited state, that is, a triplet state with a longer lifetime duration (10^−6^ s) through the process of intersystem crossing (ISC) [4,9,24]. In this triplet state, PS can undergo two different types of reactions:(1)It can react with nearby bio-substrate such as fatty acid and DNA via electron transfer to generate free radicals (anion/cation). These free radicals further interact quickly with biological molecules such as lipids, proteins, and nucleic acids to generate ROX (hydroxyl/superoxide radicals), which eventually cause cancer cell death. This is type-I PDT [9,25].(2)It can react with surrounding cellular oxygen (^3^O_2_, which has a triplet ground state) via direct energy transfer. ^3^O_2_ quickly jumps to a singlet excited state to become ^1^O_2_ (highly reactive oxygen). This is type II PDT. This singlet state toxic oxygen (^1^O_2_) can oxidize the amino acids in lipids, proteins, sugar linkages or bases in DNA and induce changes in the lipid and calcium metabolism, upregulation of stress proteins and cytokines, and ultimately, induce cell death to occur via necrosis and apoptosis [26,27,28].

## 3. Photosensitizer (PS), a Light-Activable Drug

In spectroscopy language, a compound that has a stable electronic configuration and has a ground singlet state containing a photo trigger π electron is called a photosensitizer. When a specific wavelength of light interacts with this type of compound, it gets excited into an unstable singlet state (short life), and to regain a stable state (ground state) it jumps to a triplet unstable excited state (long life), which results in complex chemical interactions with a biological substrate to generate toxic effects, that is, it is photosensitized. A photosensitizer (PS), a light-activated drug is also called a chromophore, which has unsaturated conjugated bonds to absorb a specific wavelength of light to become excited [16]. The selection of the PS in PDT is a very crucial step [7,29].

### 3.1. Features of Ideal PS

Theoretical efforts to define the characteristics of PS for human cancer have identified the following factors:(1)Should be a pure chemical compound;(2)Must have a high quantum yield (Φ_Δ_) of singlet oxygen;(3)Non-toxic effect for normal healthy cell;(4)Selective long localization period in malignant cells and fast excretion rate from healthy cells;(5)Should be integrative and have the ability to be used in combination with other therapeutic modalities such as chemo, radiotherapy, and surgery [12,29,30].

### 3.2. Conventionally Approved Photosensitizers for Cancer PDT

Currently, there are more than 400 chemical compounds that are identified as photosensitizers [7,16]. Nevertheless, only a few have been approved for clinical application in PDT, others were clinically evaluated but failed, and some are still ongoing for clinical approval [31]. The conventional PSs that have been approved for various human cancer treatments are tabulated in Table 1.

As elucidated in Table 1, few compounds have been clinically approved for PS use in PDT. Hence, researchers are attempting to discover novel and more effective PSs for the improvement of PDT. Due to the side effects of synthetic drugs, biomedical researchers have become increasingly interested in natural drugs that can be extracted from natural resources. Nature is an economical pharmaceutical factory that can be utilized to discover many new environmentally sustainable drugs.

## 4. Herbal Photo-Activable Compounds from Natural Reservoirs

In 2011, a research group reviewed the role of natural compounds in pharmaceutical discovery from 2005 to 2010 and concluded that there were 19 drugs, that have been approved worldwide. These drugs were derived from natural products [37]. To date, many plant compounds have been reported for their anticancer activity and these compounds are very cogent signs of the progress in pharmaceutically effective drugs [38,39,40,41,42,43,44]. The EM spectrum (γ-rays to radio waves) of light has a wide range of wavelength and energies. The region of the EM spectrum from 400 nm to 700 nm is not significantly different from the remaining part of the EM spectrum, except that the human eye can detect this range (visible region). Plants contain chromophores, which have an unsaturated conjugated bond. When light interacts with these molecules, the moietic nature of chromophore causes the conformational changes in it [45,46,47]. This review provides an overview of the anticancer and phototoxic activity of plant-based natural compounds that could be isolated from plants. These compounds are furanocoumarins, alkaloids, poly-acetylenes and thiophenes, curcumins, flavonoids, anthraquinones, and natural extracts.

### 4.1. Furanocoumarins

The phytochemical furanocoumarins are secondary metabolites that belong to the group of phenolic compounds consisting of coumarins. Depending upon their scaffold they are sub-classified into two groups [48] as follows:(1)The linear (psoralens) group, which includes psoralen, bergapten, and xanthotoxin;and(2)The angular (angelicins) group, which includes angelicin, pimpinellin and sphondin.

The scaffold structure of these types is elucidated in Figure 2. Psoralen is present in numerous plant families in high concentration, including Moraceae, Leguminosae, Apiaceae, and Rutaceae. Angelicin is only present in Leguminosae and Apiaceae. Different types of techniques have been used to extract, separate and analyze the furanocoumarin from natural plants, such as solid-phase extraction, supercritical fluid extraction, various chromatography techniques, e.g., thin-layer chromatography (TLC), high-performance liquid chromatography (HPLC), HPLC-mass spectrometry (HPLC-MS), gas chromatography (GC), capillary electrophoresis and pressurized capillary electro chromatography. Upon UV light exposure, furanocoumarins become toxic and generate photo dermatitis that produces toxicity and mutagenic disorders. They bind with cellular substrates such as lipids and proteins, leading to the generation of ROS and the formation of novel antigens by modification of the covalent bonding of proteins [49,50,51,52,53,54,55,56].

Fig extracts, which have high concentrations of furanocoumarins were analyzed against in vitro melanoma cell lines (C32) in conjunction with UV light. Upon UV exposure, the cytotoxic effect was enhanced [57]. Psoralen-UVA (PUVA) therapy is the main indication for many diseases such as psoriasis, atopic eczema, cutaneous T-cell lymphoma (CTCL), photodermatoses, and vitiligo. Researchers have reported that furanocoumarins have anticancer activity against skin, leukemia, and breast in vitro cell lines [50,58,59,60,61,62,63,64]. Furanocoumarins activate various cell death pathways, which lead to cancer death. In 2012, a group of researchers suggested that PUVA therapy might be applied in psoriasis through the stimulation of apoptosis (especially lymphocytes) by the suppression of keratinocytes and Bcl-2 expression via the Fas and P53 pathways, which results in the healing of psoriasis [64]. Xia and colleagues reported on the photoactivation of psoralen-UVA (PUVA) in human breast cancer. Normally upon exposure to UV, psoralen triggers the DNA inter-strand crosslinks (ICL) that prevent DNA transcription and replication. This study showed that PUVA also minimizes the p85ErbB2 phosphorylation in ErbB2+ breast cancers, which leads to the apoptosis of tumor cells [65]. Natural furanocoumarins (bergamottin (BGM)) extracted from grapefruit juice induce apoptosis to destroy tumor cells through inhibition of the signal transducer and activator of the transcription 3 (STAT3) pathway. The study predicted that BGM is a new inhibitor of the STAT3 pathway, which might prove to be an efficient strategy in multiple myeloma cells (MM) and other cancers [66]. The efficacy of PUVA has been studied in an in vitro model of cutaneous T-cell lymphoma (CTCL) cell lines (that have either functional p53 or inactive functional p53); in both cases, apoptosis occurred along with the upregulation of the mitochondrial genes BAK, BAX, and PUMA. There was also a downregulation in the BCL-2. Type I interferons (IFNs) were suggested via the JAK1 pathway instead of P53-induced apoptosis. This p53-independent apoptosis was induced by PUVA [67].

In brief, it is concluded that furanocoumarins activate several pathways to induce cell death, such as inhibition of phosphatidylinositol-3-kinase, STAT3, AKT protein expression, and nuclear factor-κB (NF-κB) [53,54,56,65,66,67,68,69,70], as shown in Figure 3.

### 4.2. Alkaloids

Alkaloids are a group of secondary metabolites that contain cyclic structures (organic compounds) with nitrogen atoms. These organic compounds are basically naturally photoactive, and primarily found in plants, especially flowering plants. Many higher plants also contain alkaloids. Based on their carbon scaffold, alkaloids are classified as indoles, pyridines, quinolines, isoquinolines, pyrrolidines, pyrrolizidines, tropanes, steroids, and terpenoids [71,72,73,74,75]. Alkaloids have been reported to have various effective properties, such as analgesic, neuropharmacologic, local anesthetic and pain relief, neuropharmacologic, antifungal, antimicrobial, anti-inflammatory, anticancer, antifungal, and numerous other activities. They are also beneficial as supplements, diet ingredients, pharmaceuticals (such as camptothecin and vinblastine, which are used as chemo drugs [76]), and in other applications in human life [77,78,79,80,81,82]. Berberine (an alkaloid), which is associated with low-density lipoproteins (LDL), successfully accumulates in the mitochondria of the human primary glioblastoma (U87-MG) cell line and upon irradiation it shows phototoxicity, leading to apoptosis [37]. Coralyne and UVA light (CUVA) triggered apoptosis in human skin (A431) cancer cells through lysosomal and mitochondrial disorder. BAX was in the silencing phase, while caspase-8 and lysosomal proteases activation led to a BID cleavage. In the apoptosis response, a twin signaling axis JAK2-STAT1 and ATR-p38 MAPK pathways worked upstream of BAX-activation. Overall, the researchers concluded that Coralyne (topoisomerase-I inhibitor) could be a remarkable photoactivated agent in the treatment of skin cancer [83]. Another study on alkaloids utilized renal carcinoma cell lines to analyze the photosensitive characteristics of berberine with PDT. Berberine-mediated PDT promotes apoptosis and autophagy through caspase 3 activity due to ROX production. Due to its notable anticancer effect on renal carcinoma cells, berberine was recommended as a photosensitizing agent that could potentially be effective in PDT [84]. Due to the antineoplastic and phototoxic properties, berberine-mediated PDT was also used against cervical carcinoma cells to evaluate its effect. The study found that there was an increase in caspase-3 activity and ROS generation, which promotes the killing of cells via caspase-dependent apoptosis [85].

### 4.3. Poly-Acetylenes and Thiophenes Derivatives

Poly-acetylenes and thiophenes derivatives are secondary compounds, whose photoactivated features in insects were first revealed by Arnason et al. in Canada (1975) [86]. Poly-acetylenes consist of conjugated double (=) and triple (≡) bonds and are bio-synthetically cyclized into thiophene compounds. Depending upon their scaffold they can be sub-classified into the following three groups:(1)Straight-chain aliphatic acetylenes;(2)Partly cyclized; and(3)Thiophene derivatives (addition of sulfur into polyacetylene) [87].

Normally conjugation of aliphatic compounds with less than three acetylenic bonds does not show a very efficient phototoxic effect in nature. Poly-acetylenes are usually less phototoxic than thiophene derivatives. Both poly-acetylenes and thiophenes derivatives have an absorbance maximum for photobiological effects in the wavelength range of 314–350 nm. These compounds have been found in many plant families such as Asteraceae, Campanulaceae, Araliaceae, Apiaceae, and Basidiomycete fungi groups. Poly-acetylenes and thiophenes derivatives have been reported to have various pharmacological actions such as antimicrobial activities, antitumor, anti-inflammatory, anticancer, and analgesic [82,83,84,85,86,87,88]. Wang et al. isolated four thiophens: 5-(4-hydroxy-1-butynyl)-2,2′-bithiophene, 5-(3,4-dihydroxybut-1-ynyl)-2,2′-bithiophene, 5-(4-hydroxybut-1-one)-2,2′-bithiophene and 5-{4-[4-(5-pent-1,3-diynylthiophene-2-yl)-but-3-yny}-2,2′-bithiophene from E. latifolius Tausch (ELT) plants (which belong to the Asteraceae family) and tested them against human cervical carcinoma (HeLa) and human malignant melanoma (A375-S2) cell lines. These compounds showed cytotoxic activity in the dark but the cytotoxicity was sufficiently increased upon ultraviolet A (UVA) irradiation for 30 min. The mode of phototoxicity of these natural thiophenes was type II PDT. These molecules accumulate in fatty acid-rich regions and produce singlet oxygen to destroy the cell membrane [87,88]. Bioactive thiophenes from the extract of Echinops grijisii (E. grijisii) show cytotoxicity against human chronic myelogenous leukemia (K562), human acute myeloid leukemia (HL-60), human colon cancer (SW620) and human hepatocarcinoma (HepG2 and MFC-7) [89,90,91,92]. Based on the above discussion, we can say that the combination of poly-acetylenes and thiophene derivatives with PDT could have the potential to improve the efficacy of treatment.

### 4.4. Curcumins

Curcumin (CU) is a polyphenol pigment with a yellow color. It is primarily found in rhizomes and roots of the turmeric plant, Curcuma longa (family Zingiberaceae, species Curcuma). Curcuminoid is one of the most studied bioactive plant-derived compounds [93,94,95].

Its absorption wavelength varies from 405 to 435 nm; therefore, blue light is used to activate curcumin. Because curcumin has a hydrophobic nature, it must be modified before it can be used as a PS in PDT applications [96,97,98,99]. Different studies have reported that curcumin is an efficient anticancer agent that inhibits cell proliferation in various types of cancer such as lung, kidney, liver cancers, breast, colon, prostate, and ovary. The extracellular matrix (ECM) plays an important role in cell growth, survival, differentiation, and motility. The enzyme matrix metalloproteases (MMPs) break down the ECM to enhance tumor invasion. Curcumin reduces the matrix metalloproteases’ (MMPs) enzyme production, chemokines, tyrosine kinase protein, inhibits N-terminal activity, the cell surface adhesive molecules NF-κβ, AP-1, TNF-α, LOX, and COX-2 and growth factors (HER-2 and EGFR) to inhibit the tumor invasion and progression. In some tumors, suppressing angiogenic cytokines (IL-6, IL-23, and IL-1β) inhibits angiogenesis [100,101,102,103,104,105,106,107]. A research study investigated the combined effect of curcumin with UVB light on human keratinocyte cells (HaCaT) and found that this combination induced apoptosis as compared to UVB or curcumin alone. In addition, UVB irradiation combined with curcumin stimulates apoptosis in HaCaT cells by activating caspase -3, -8, and 9 followed by cytochrome c release [108]. At 30 J/cm^2^, demethoxycurcumin (DMC)-PDT was also effective against MCF-7 breast cancer cells. It was found that in DMC-PDT the early event was autophagy and the late event was apoptosis [109]. Bernd concluded that the CU’s ability to produce ROS and its anticancer properties make it a promising candidate for use as a natural PS [110]. Curcumin was also effectively used in antimicrobial PDT studies [111,112,113,114].

According to the discussion above, CU can be utilized as a natural PS, and when combined with PDT, it can achieve high therapeutic efficacy at low concentrations. Based on the existing PDT and photoactive research data on CU, it appears that CU could be a promising natural PS for PDT [115,116]. CU has the potential to be used an effective photosensitizer in the treatment of cancer and skin infections. Thus, further research regarding the photodynamic potential of CU derivatives in terms of enhanced absorption and the extinction coefficient will help to improve photodynamic toxicity efficacy.

### 4.5. Flavonoids

Flavonoids are secondary metabolites; this group of polyphenolic compounds consists primarily of a benzopyrone structure. They are commonly found in vegetables, cereals, herbs, flowers, fruits, seeds, and nuts and are responsible for many pharmacological activities [117,118]. Previous research has shown that these compounds are used to treat different types of diseases such as coronary heart disease prevention and have properties such as anti-inflammatory, anticancer, antiviral, antioxidant, and hepatoprotective [119,120]. Depending upon their degree of saturation, oxidation of the carbon ring and chemical structure, flavonoids are classified into the following seven subgroups [121,122]: (1) flavanones, (2) flavones, (3) flavonols, (4) isoflavones, (5) anthoxanthins (flavanone and flavanol), (6) anthocyanins, and (7) chalchones.

In 2014, a research group described the photosensitizing effect of flavonols (quercetin) on PDT of larynx (Hep-2) cancer cells [122]. The absorption wavelength of quercetin is 557 nm. The quercetin (50 µM) enhanced the therapeutic outcomes of aluminum phthalocyanine tetrasulfonate (10 M), mediated PDT-induced cytotoxicity, and resulted in a reduction in mitochondrial membrane potential and apoptosis in cancer cells [122]. Similarly, the PDT effect was enhanced in human thyroid anaplastic (SNU 80) cells in the presence of genistein, whose absorption wavelength is 382 nm. A dose of 25 µM genistein enhanced the photofrin-mediated PDT by increasing ROS levels complemented by modification in the expression of apoptosis-related proteins, cytochrome c, Caspase 3, Caspase 9, Caspase 8, and Caspase 12. The efficacy of stimulating apoptosis in the SNU 80 cell line was enhanced by the genistein–photofrin-mediated PDT [123]. The combination of genistein (IC50-50 µM) with hypericin (IC50-0.021 µM) stimulated photoactivated apoptosis by decreasing Bcl-2 protein [124]. In another study, the genistein-5-aminolevulinic acid-mediated PDT increased ROS and DNA damage, inhibited cell proliferation, and eventually enhanced apoptosis in leukemia cells [125]. Chrysanthemin, also known as cyanidin-3-glucoside, is a kind of anthocyanin that has an absorption wavelength of 550 nm, which has been found to have photoactivable characteristics. In 2017, it was used in antimicrobial PDT against the periodontopathic bacterium Porphyromonas gingivalis using a green light laser in an in vitro biofilm model. It was observed that cyanidin-3-glucoside (404 µM) efficiently reduced Porphyromonas gingivalis in biofilm [126].

Riboflavin has also been used in antimicrobial PDT because it has two absorption wavelengths at 360 and 440 nm [102]. Riboflavin (100 M) inactivates Listeria monocytogenes on smoked salmon fillets after 90 min of exposure to 460 nm wavelength of light (150–580 W/m^2^) [127].

### 4.6. Anthraquinones

Anthraquinone (AQ) is the most common type of natural quinone found in higher plants. The shikimic acid/mevalonate and acetate/malonate routes yield anthraquinones, which are generally categorized as monomeric and dimeric anthraquinones [128]. Representative AQs include rubiadin (410 nm excitation wavelength), emodin (434 nm), rhein (437 nm), physcion (438 nm), carminic acid (494 nm), and purpurin (515 nm), which are depicted in Figure 4.

In 2011, Comini et al. showed that irradiating AQs (rubiadin and soranjidiol) with visible light at 380–480 nm can enhance the antiproliferative activity of the MCF-7 breast cancer cell line [129].

Another research study was conducted in 2021, both in vitro and in vivo in a model of subcutaneously implanted tumors to investigate the photosensitizing capacity of PTN in breast cancer LM2 cell lines. Parietin (PTN, 30 µM)-mediated PDT (1 h and blue light of 1.78 J/cm^2^) produced a considerable reduction in the cell migration of LM2 cells. This suggested that it could be an inhibitor of metastasis. Fluorescence microscopy revealed that the AQ was localized in the cytoplasm, with no fluorescence detected in the nucleus. In an in vivo study, the tumor histology demonstrated extensive tumor necrosis up to a depth of 4 mm [130].

Vittar et al. have observed the photosensitization effects of AQs in Balb/c mice as well as their dose-dependent leukocyte-inhibiting abilities via triggering autophagy, necrosis, and apoptosis [131]. Based on these results, we can conclude that the photoactive nature of AQs could suppress cancer cell proliferation.

### 4.7. Natural Extracts

In addition to single chemicals from natural reservoirs, natural extracts have also been employed as PSs in a number of investigations. Plant extracts with anticancer and photosensitizing properties can be used as potential photosensitizers in PDT.

The photosensitizing effect of phthalocyanine in breast cancer has been shown to be enhanced by root extracts of the berry variety, Rubus fairholmianus. In comparison to conventional photosensitizers, R. fairholmianus root extracts displayed phototoxicity, and the recombination of extracts with PDT induced caspase-mediated death [132]. In another study, Villacorta et al. found that in breast cancer cells (MCF7), the extracts of Albizia procera and Lumnitzera racemosa showed enhanced cytotoxicity by natural photosensitizers in PDT. The crude extracts were shown to be non-toxic to cancer cells however, the photoactivated extracts displayed cytotoxicity against MCF7 when illuminated with 5.53 mW white light [43]. Liao et al. hypothesized that two Chinese medicinal plants, Rhizoma coptidis, and Cortex phellodendri, might be employed as photosensitizers based on their fluorescence characteristics. When the extracts were combined with light, they caused cell death as well as reduced cell viability and proliferation [133]. In a clinical trial, Gonçalves et al. investigated the effects of a Bixa orellana extract on Gram-negative biofilms and conducted clinical investigations. They reported that 20 s of exposure to a dose of 1540 W/m^2^ light with the extract reduced halitosis [134]. Chlorophyllin, curcumin, tolyporphin hypericin, and many other individual phytocompounds, as discussed above, are also present in plant extracts. These compounds could contribute to the photosensitizing activity of certain plant extracts [135].

The molecular pathways responsible for natural extracts and the anticancer actions of key phytochemicals are summarized in Figure 3, which is based on prior studies and the aforementioned studies. With respect to the future of the strategy, the use of photodynamic therapy for large or deep-seated tumors is an area of unmet clinical need that is rarely considered in preclinical studies. Light delivery using multiple interstitial compounds, as well as nano-delivery protocols [136], is a promising approach that merits further research. Considerable room exists for the clinical expansion of new PDT platforms with technological innovations and strategic advancements.

## 5. Conclusions

In this review article, we discussed PDT, conventionally approved photosensitizers (PSs), and the scope of plant-based photoactive compounds, which can be used as a natural PS to overcome the side effects of synthetic drugs. Many natural compounds have cytotoxicity in the dark for cancer cells but upon light exposure, their toxicity is extensively enhanced and different cell death pathways are activated. Natural compounds have very complex chemistry, which still needs to be explored. The photoactive properties of many natural compounds are still unexplored and await investigation to enhance the use of natural compounds in pharmaceutical research. Numerous studies have used broad-spectrum light to evaluate the phototoxic activity of natural compounds. The obtained results could be more significant if monochromatic light was employed. Exploring new natural compounds such as PS will help to enhance the efficacy of PDT and also reduce side effects in comparison with conventional PS.

As biomedical technologies continue to produce light sources with increased power, the interest in cancer phototherapies is expected to remain high. In fact, phototherapies are actively being pursued for a broad range of indications. Moreover, nanoscale photosensitizing agents have produced impressive preclinical results, but with limited clinical translation thus far. The advanced targeting and activation features of these agents might lead to manufacturing complexities that impede their clinical translation. Nonetheless, photoimmunotherapy and antibody-targeted PDT are currently under investigation in clinical trials and have the potential to become a next-generation therapeutic protocol.

## Figures and Tables

**Figure 1 biomedicines-11-00224-f001:**
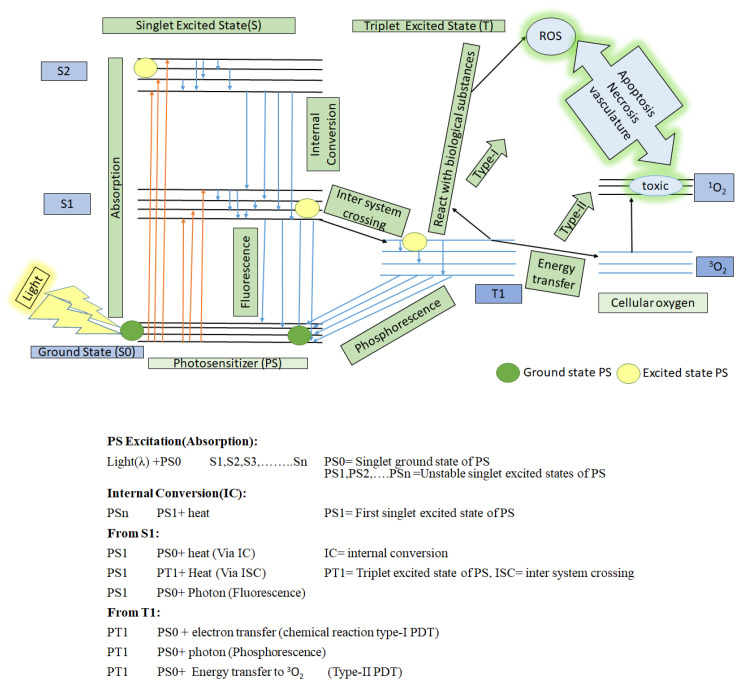
A schematic representation of photosensitization that includes type-I and type-II PDT on the basis of the Jablonski diagram. Redrawn from [4].

**Figure 2 biomedicines-11-00224-f002:**
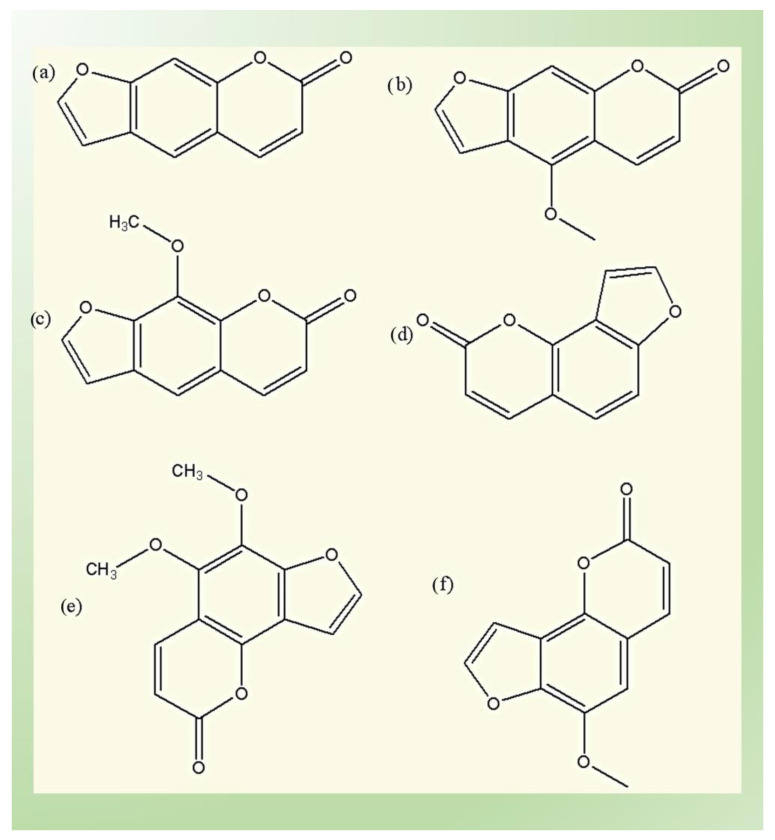
Linear and angular groups of furanocoumarins: (**a**) psoralen; (**b**) bergapten; (**c**) methoxsalen; (**d**) angelicin; (**e**) pimpinellin; and (**f**) sphondin.

**Figure 3 biomedicines-11-00224-f003:**
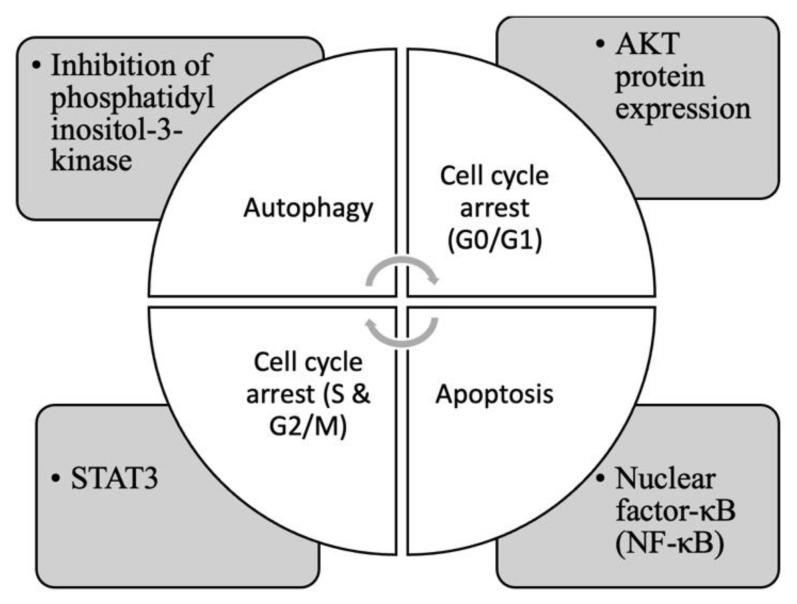
Common pathways that induce cancer cell death by furanocoumarins.

**Figure 4 biomedicines-11-00224-f004:**
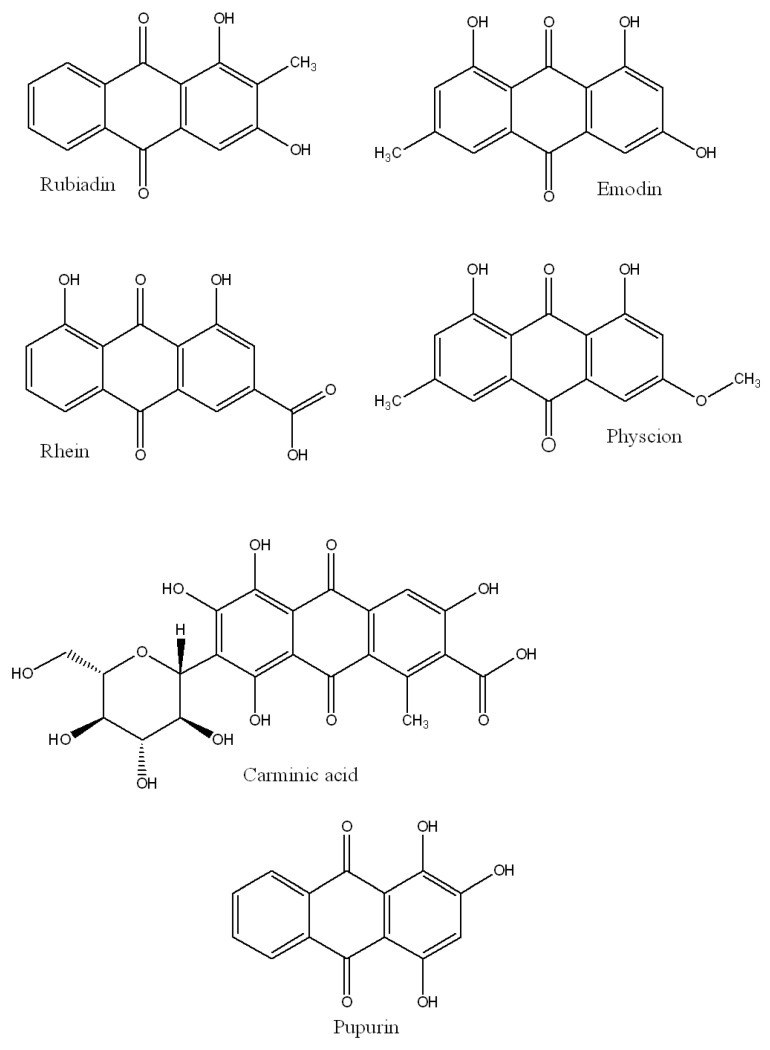
Chemical scaffold of anthraquinones.

**Table 1 biomedicines-11-00224-t001:** Conventionally approved PS employed in PDT for cancer treatment.

Photosensitizer	Generic Name	λ (nm) Max.	Chemical Structure	Drug Light Interval	Approved for	Ref.
5-Aminolevulinic acid (ALA)	Luvalan	635	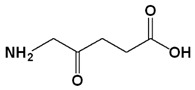	6 h	Actinic keratosis (USA 1999)	[32,33]
Hematoporphyrin derivatives (HPD)	Photofrin (Porfmer sodium)	630	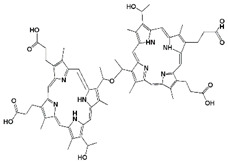	48 h	(i) Bladder cancer (Canada 1993) (ii) Early-stage lung cancer (Japan 1994) (iii) Esophageal cancer (FDA USA 1995), Early-stage non-small-cell lung cancer (FDA USA 1998)	[29,31]
Meta-tetra(hydroxyphenyl) chlorin (mTHPC)	Foscan (temoporfin)	652	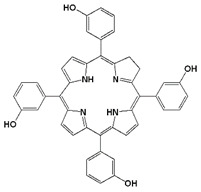	96 h	(i) Head and neck squamous cell carcinoma (Europe 2001)	[29,31]
Benzoporphyrin derivative monoacid ring A	Verteporfin or Visudyne	690	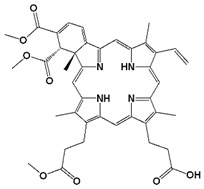	30 min	Choroidal neovascularization (age-related macular degeneration (AMD) (FDA 2000))	[34]
Palladium (Pd)—substituted bacteriochlorophyll derivative	Tookad (WST09 (padoporfin) WST11 (padeliporfin or TOOKAD Soluble))	763	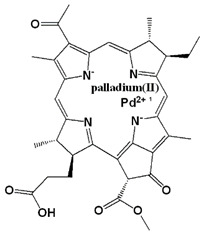	Short interval mins	Clinical trial for prostate cancer	[29,35,36]
N-aspartyl chlorine e6 (NPe6)	Talaporfin sodium (Laserphyrin^®^)	664	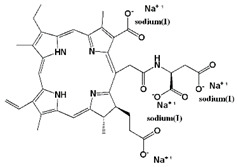	0.25–4 h	(i) Early-stage lung cancer (Japan 2003)	[12,29]

## Data Availability

Not applicable.

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
