# Peer review of "An Overview of Potential Natural Photosensitizers in Cancer Photodynamic Therapy"

_biomedicines, 2023, doi:10.3390/biomedicines11010224_

Round 1

Reviewer 1 Report

Overall, the quality of the manuscript is high, and in my opinion it can be accepted after minor revisions.

Check the language of the manuscript as there were some grammatical mistakes inthe manuscript as well as some misspelling errors. 

The future trends of the current work should be highlighted before the conclusion section 

Please prepare a brief graphical abstract for the curernt manuscript as it will become more attractive to the readers and it summarizes the whole manuscript.

Author Response

Dear Reviewers;

Re: [Manuscript Title: "An overview of potential natural photosensitizers in cancer photodynamic therapy"]

Please find our responses to your comments / suggestions below:

Reviewer 1:

Comments and Suggestions for Authors

1- Overall, the quality of the manuscript is high, and in my opinion it can be accepted after minor revisions.

            We appreciate the encouraging comment of this Reviewer and thank him/her for his/her valuable suggestions.

2- Check the language of the manuscript as there were some grammatical mistakes in the manuscript as well as some misspelling errors.

            The language of the entire manuscript has been checked by a Professional Editing and Proof-Reading service.

3- The future trends of the current work should be highlighted before the conclusion section

            A new paragraph has been added before the conclusion section as suggested.

4- Please prepare a brief graphical abstract for the current manuscript as it will become more attractive to the readers and it summarizes the whole manuscript.

            A Graphical Abstract has been prepared and will be submitted along with the revised manuscript as per suggestion of this reviewer.

Reviewer 2 Report

Comments on the manuscript were as follows

Comments and Suggestions for Authors

The article “An overview of potential natural photosensitizers in cancer photodynamic therapy” by Aziz at al. is of good quality and clear. In this review article, the potential of phytochemicals to act as natural photosensitizers for PDT and their mode of action are discussed in detail. I recommend this paper to be published in the journal. Here are some suggestions:

1: The “Introduction” of the manuscript should be more concise.

2: As a review article, to be complete, the authors should enrich the related articles published in 2022 (such as doi: 10.3390/molecules27041192; doi: 10.1016/j.pdpdt.2022.102801; doi: 10.1021/acsabm.2c00228.) In fact, this manuscript does not cite one literature published in 2022, which is not allowed.

3: Unnecessary literatures (References 53-58, 124-137, and 145-150) should be removed.

4: In lines 67-68, “Since very ancient times, herbal medicine for treating various human cancerous and non-cancerous diseases was very common.”. The authors should enrich the related articles. “Since very ancient times, herbal medicine for treating various human cancerous (doi: 10.3390/biomedicines9060689; doi: 10.3389/fphar.2022.1036502) and non-cancerous (doi: 10.3389/fphar.2022.926507; doi: 10.3390/biomedicines9050472) diseases was very common.”.

5: The “Conclusions” of the manuscript, drawbacks, challenges and possible solutions of PDT should be discussed.

Author Response

Reviewer 2:

Comments and Suggestions for Authors

Comments on the manuscript were as follows

The article “An overview of potential natural photosensitizers in cancer photodynamic therapy” by Aziz at al. is of good quality and clear. In this review article, the potential of phytochemicals to act as natural photosensitizers for PDT and their mode of action are discussed in detail. I recommend this paper to be published in the journal. Here are some suggestions:

            We appreciate the instructive comment of this Reviewer and thank him/her for his/her valuable time and suggestions.

1: The “Introduction” of the manuscript should be more concise.

            As per suggestion of Reviewer 1, a Professional Editing and Proof-Reading service was employed to double-check the entire Manuscript. Considering that the INTRODUCTION is only 557 words, any alteration to this section will adversely affects the message tried to be conveyed to readers.  

2: As a review article, to be complete, the authors should enrich the related articles published in 2022 (such as doi: 10.3390/molecules27041192; doi: 10.1016/j.pdpdt.2022.102801; doi: 10.1021/acsabm.2c00228.) In fact, this manuscript does not cite one literature published in 2022, which is not allowed.

            Two of the 3 suggested references, relevant to the manuscript, are incorporated.

3: Unnecessary literatures (References 53-58, 124-137, and 145-150) should be removed.

            All unnecessary References have been removed as suggested by this Reviewer.

4: In lines 67-68, “Since very ancient times, herbal medicine for treating various human cancerous and non-cancerous diseases was very common.”. The authors should enrich the related articles. “Since very ancient times, herbal medicine for treating various human cancerous (doi: 10.3390/biomedicines9060689; doi: 10.3389/fphar.2022.1036502) and non-cancerous (doi: 10.3389/fphar.2022.926507; doi: 10.3390/biomedicines9050472) diseases was very common.”.

            The suggested 4 References have been incorporated in the manuscript as suggested.

5: The “Conclusions” of the manuscript, drawbacks, challenges and possible solutions of PDT should be discussed.

            As suggested, a new paragraph is added at the end of the Conclusions.

Sincerely

Prof. Dr. M. R. Mozafari (Corresponding Author)
